# The Impact of a Dedicated Sedation Team on the Incidence of Complications in Pediatric Procedural Analgosedation

**DOI:** 10.3390/children9070998

**Published:** 2022-07-02

**Authors:** Sofia Apostolidou, Mirna Kintscher, Gerhard Schön, Chinedu Ulrich Ebenebe, Hans-Jürgen Bartz, Dominique Singer, Christian Zöllner, Katharina Röher

**Affiliations:** 1Division of Neonatology and Pediatric Intensive Care Medicine, Department of Pediatrics, University Medical Center Hamburg-Eppendorf, D-20246 Hamburg, Germany; m.kintscher@uke.de (M.K.); c.ebenebe@uke.de (C.U.E.); d.singer@uke.de (D.S.); 2Institute of Medical Biometry and Epidemiology, University Medical Center Hamburg-Eppendorf, D-20246 Hamburg, Germany; g.schoen@uke.de; 3Office for Quality Management and Clinical Process Management, University Medical Center Hamburg-Eppendorf, D-20246 Hamburg, Germany; h.bartz@uke.de; 4Center of Anesthesiology and Intensive Care Medicine, Department of Anesthesiology, University Medical Center Hamburg-Eppendorf, D-20246 Hamburg, Germany; c.zoellner@uke.de (C.Z.); k.roeher@uke.de (K.R.)

**Keywords:** children, sedation, analgesia, propofol, adverse events

## Abstract

The number of pediatric procedural sedations for diagnostic and minor therapeutic procedures performed outside the operating room has increased. Therefore, we established a specialized interdisciplinary team of pediatric anesthesiologists and intensivists (Children’s Analgosedation Team, CAST) at our tertiary-care university hospital and retrospectively analyzed the first year after implementation of the CAST. Within one year, 784 procedural sedations were performed by the CAST; 12.2% of the patients were infants <1 year, 41.9% of the patients were classified as American Society of Anesthesiologists (ASA) grade III or IV. Most children received propofol (79%) and, for painful procedures, additional esketamine (48%). Adverse events occurred in 51 patients (6.5%), with a lack of professional experience (OR 0.60; 95% CI 0.42–0.81) and increased propofol dosage (OR 1.33; 95% CI 1.17–1.55) being significant predictors. Overall, the CAST enabled safe and effective procedural sedation in children outside the operating room.

## 1. Introduction

Pediatric analgosedation is steadily developing to a highly specialized anesthesiologic and pediatric intensivist service for a growing number of procedures. The first established sedation guidelines were published by the American Academy of Pediatrics (AAP) in 1985 [1], followed by the American Society of Anesthesiology (ASA) in 2002 [2]. In the meantime, a wide range of drugs and techniques for use in pediatric sedation developed, resulting in a significant variance of sedation levels, effectiveness, and associated risks [3,4,5,6]. There is growing evidence for the need for deep sedation for many pediatric procedures [7]. However, there is no standardized recommendation on which medication to choose for extended and painful procedures [8,9]. The need for guidelines specifying safety precautions to minimize the incidence of adverse events is increasingly claimed [7,10,11]. Therefore, we established a specialized interdisciplinary team of pediatric anesthesiologists and pediatric intensivists at our tertiary-care university hospital called Children’s Analgosedation Team (CAST) which performs all procedural sedations in children outside the operating room. In the previous studies investigating adverse events in pediatric procedural sedation, the sedations were performed in settings not involving a specialized sedation team [9,12]. The primary aim of the present study was to analyze the incidence of adverse events for procedural sedation conducted by a dedicated interdisciplinary sedation team. Furthermore, the study aimed to identify potential risk factors for adverse events.

## 2. Methods

This retrospective observational study was approved by the local ethics committee, which waived informed consent. We reviewed the medical records of all children receiving procedural sedation from August 2014 to August 2015. Several children received multiple sedations during the reviewed period. All the sedations were performed by the interdisciplinary CAST, which is staffed by four anesthesiologists and two pediatric intensivists and provides procedural sedation for children from 0 to 18 years of age. The sedations were performed outside the operating room: at the radiology department, hospital wards, and outpatient departments. According to the standard clinical management of the CAST, propofol and midazolam were available as sedative agents and esketamine or remifentanil were used as optional adjunctive analgesic drugs. Standard monitoring consisted of oxygen saturation, heart rate, noninvasive blood pressure, and capnography. All the patients received supplemental oxygen during sedation. Demographic and clinically relevant data were recorded for each sedation. An upper respiratory tract infection was defined as the presence of a runny nose or cough. Adverse events (AEs) were captured from the beginning of drug administration until the patient was transferred to the ward, the outpatient department, or the perioperative anesthesia care unit (PACU). AEs were categorized in respiratory, hemodynamic, and other adverse events. Aspiration, vomiting/regurgitation, desaturation < 90% for > 30 s, hypotension < 50% of baseline, laryngospasm, thorax rigidity, unplanned admission to the pediatric intensive care unit, cardiac arrest, and death were categorized as serious adverse events (SAEs). All the documented anesthesiologic interventions during the sedation procedure were recorded.

## 3. Statistical Analysis

All the analyses were conducted using the R version 4.0.3 software. Descriptive data are expressed as the medians and range for continuous variables and as counts and category percentages for categorical variables. The primary outcome variable of interest was the occurrence of at least one AE. The independent variables of interest were age, sex, ASA grade, date of sedation, category of the primary diagnosis, type and dose of a sedative, use of an analgesic, and presence of upper respiratory tract infection. Age was used as a continuous and categorical variable with the following categories: one-year-old or younger, older than one-year-old to six years old, and older than six years old. The sedation dates were grouped by a three-month interval and served as a surrogate for the team’s experience. In the first step, a bivariate analysis for all the independent variables was conducted. We used a random coefficient model for considering a cluster effect as some patients received several sedations. In the second step, all the significant independent variables were entered into an analysis of variance using type II Wald chi-squared tests. The model was adjusted for age and the ASA classification. Odds ratios and 95% confidence limits were computed for each independent variable, and a *p*-value of less than 0.05 was considered statistically significant.

## 4. Results

During one year, the CAST provided 792 sedations. Eight sedations were excluded from the analysis because of missing medical records. So, 784 sedations performed in 442 children were eligible for analysis. The median age was 5.3 years (range 2 days–20 years). 12.2% of the patients were infants younger than one year, and 41.9% of the patients were ASA grade III or IV. All demographic characteristics are presented in Table 1. The most common category for the patient’s primary diagnosis was hematology/oncology (415 sedations, 52.9%), followed by neurology (132 sedations, 16.8%), hepatology (93, 11.9%), and nephrology (56, 7.1%), with all other categories being less frequent. Upper respiratory tract infection was present in 41 patients (5%). Of all the procedures, 58% were painful (Table 2). In 79.1% of the sedations, the patients received propofol either as bolus administration alone or as a bolus followed by continuous infusion. The median dose of propofol bolus for induction of sedation was 3.3 mg kg^−1^ (range, 0.5–17 mg kg^−1^), and the median dose of continuous propofol infusion was 6.9 mg kg^−1^ h^−1^ (range: 1–14 mg^−1^ kg^−1^ h). Midazolam was applied in 17% of the sedations with a median dose of 0.15 mg kg^−1^ (range: 0.02–0.7 mg kg^−1^). In 57% of the sedations, the patients received an adjunctive analgesic drug. The most frequently used combination was propofol and esketamine (57%), followed by midazolam and esketamine (29%) and propofol and remifentanil (14%). For esketamine and remifentanil, the median administered doses were 1.1 mg kg^−1^ and 0.15 µg kg^−1^ min^−1^, respectively.

The overall incidence rate of AEs was 6.5% (51 procedures with AEs). Most AEs were categorized as respiratory (4.2%), whereas 2.3% were assigned to the category “Other”, and only one AE—to the category “Hemodynamic”. The most frequent AE was apnea (1.7%). SAEs were documented in only seven cases (0.9%), including six cases of desaturation < 90% for > 30 s and one case of thorax rigidity (Table 3). All the patients with SAEs had a syndromic disease. In two cases, the airway had to be secured with a laryngeal mask or intubation. The other four patients recovered rapidly after intervention with the jaw thrust maneuver, nasopharyngeal airway, and head repositioning. Pronounced thorax rigidity in one patient occurred after the application of remifentanil. There were 56 interventions during 43 sedations (5.5%), some with multiple interventions. The most frequent interventions were airway interventions, like bag-mask ventilation (2.2%), the use of a nasopharyngeal airway (1.4%), and suction of secretions (1.1%) (Table 4). Four (0.5%) procedures had to be stopped due to an AE.

The incidence of AEs was highest in infants younger than one year (14.9%) compared to children older than one year to six years or older than six years (6.4% and 5.1%, respectively). In the bivariate analysis, the risk of AEs was significantly lower in children older than one year to six years or children older than six years compared to children younger than one year (odds ratio (95% CI): 0.303 (0.105–0.875) and 0.200 (0.064–0.624), respectively; Table 5). The analysis of variance revealed a significant reduction of AEs for each quarter of increasing experience of the CAST. In contrast, female sex and each increase of propofol bolus by 1 mg kg^−1^ were independent risk factors for AEs (odds ratio (95% CI): 1.339 (1.183–1.550) and 1.331 (1.172–1.546), respectively; Figure 1).

## 5. Discussion

In the past, pediatric analgosedation for diagnostic and therapeutic treatment was a neglected procedure in our institution, often performed by a pediatrician with no specialized skills in intensive care medicine, no support from an intensive care nurse, no documentation neither about the selection and dosage of analgetic or sedative agents nor of any adverse events occurring during the sedation. With the growing demands of quality management and quality assessment, a new specialized team called CAST has been established to perform these procedural sedations. In the absence of eligible data prior to implementation of the CAST, we cannot draw any conclusions from the past procedural sedation practice, thus limiting the generalizability of this project.

The primary aim of our study was to investigate the incidence of adverse events for procedural sedation conducted by a dedicated interdisciplinary sedation team. We found an overall rate of AEs and SAEs of 6.5% and 0.9%, respectively. Most AEs were classified as respiratory rather than hemodynamic, with apnea being the most frequent. In one of the largest prospectively collected datasets compiled by Cravero et al., the overall rate of complications during pediatric procedural sedation amounts to 6%, corresponding to our results [10]. In other studies, partly including large cohorts of adult patients, the AE rates were distinctly higher [13,14]. In the pediatric population, respiratory AEs are more frequent than hemodynamic or other AEs [15], probably due to small anatomic proportions and limited respiratory reserves. Furthermore, AEs during analgosedation without securing the airway are typically respiratory in nature [10]. The rate of serious adverse events in our study was less than 1%. Unplanned serious airway intervention was necessary in 0.5% of the cases. All the adverse events were resolved by the sedation team themselves. The low rate of serious adverse events probably resulted from competent management of minor adverse events like apnea needing bag-mask ventilation, thus eliminating problems leading to a worsening clinical condition of the patient.

Our secondary aim was to identify potential risk factors of AEs for procedural sedation performed by a dedicated interdisciplinary sedation team. The present study revealed each quarter year of increasing experience of the CAST to reduce the risk for AE significantly. Furthermore, we found female sex and each increase of propofol bolus by 1 mg kg^−1^ to be independent risk factors for AEs. Within the first six months after implementation of the CAST, the number of adverse events during procedural sedation dropped by half. This might be explained by the fact that our team was small and consisted of well-equipped and well-trained nurses and physicians dedicated to sedation, which are optimal preconditions to achieve a rapid acquisition of experience within the team and a significant reduction of procedural failures [16,17]. Coté et al. demonstrated that professionals who lack adequate sedation competence are a significant risk factor for the occurrence of major complications rather than the pharmacological characteristics of applied drugs [18,19]. The fact that standardized sedation algorithms seem to account for more safety should initiate definitions of the required qualification and training for the staff performing the sedation and the procedure [20].

In our study, the sedative drug used most was propofol (nearly 80% of the cases). The most frequent combination for painful procedures was propofol and esketamine (57% of the cases). We detected a higher dosage of propofol as an independent risk factor for the occurrence of an adverse event. A recent prospective cohort study by the Canadian Sedation Safety Study Group reported similar results, with the highest observed incidence of serious adverse events for propofol alone or the combination of propofol with ketamine [21]. Propofol is considered an extremely safe and efficient sedation drug regarding procedural success rate, patient recovery time, and physician satisfaction [22]. Therefore, it can be highly recommended as the first-line sedative drug [23]. In our study, adverse events potentially related to propofol application like apnea (1.7%) and oxygen desaturation requiring intervention (0.8%) were rare. The team’s growing experience was related to a significant reduction of adverse events, probably due to the increased competence and routine use of sedation medication in a non-general anesthesia setting. Almost half of our patients were children with ASA grade III or IV. Prior published literature suggests higher ASA grade as an independent risk factor for the occurrence of adverse events [24]. Our results could not identify a higher ASA grade as a significant risk factor for adverse events. Nevertheless, ASA grade III or greater should be considered for a general anesthesia procedure [12]. The ASA classification as a predictor of the outcome in these procedures is controversial as it does not entirely reflect the actual patient’s clinical condition. However, it can be utilized as one component of the patient’s pre-sedation clinical evaluation.

There are several limitations to our analysis. The study design is retrospective and is, therefore, dependent on accurate medical documentation. For example, not all interventions during sedation could be attributed to a specific complication. Procedure-dependent risk factors (procedure type and provider type) were not analyzed in our study but are relevant factors in further risk stratification and optimizing outcomes in high-performing centers such as ours.

## 6. Conclusions

The introduction of an interdisciplinary team dedicated to pediatric sedation enabled safe and effective procedural sedation outside the operating room. The risk of adverse events significantly decreased with the growing experience of the team.

## Figures and Tables

**Figure 1 children-09-00998-f001:**
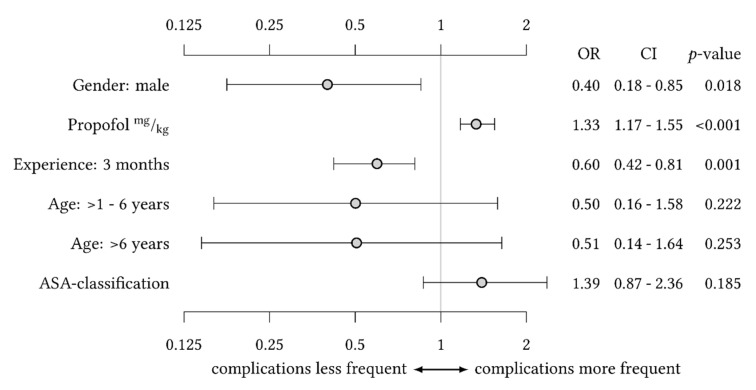
Analysis of variance for the risk of adverse events with gender, sedative agent, and quarter of team experience as independent variables of significant influence and adjustment for age and ASA classification.

**Table 1 children-09-00998-t001:** Demographic characteristics of 442 patients.

Age (years)	5.33 (0–20)
<1 year	54 (12.2)
1–6 years	190 (43.0)
>6 years	198 (44.8)
Weight (kg)	20 (2–145)
Male	254 (57.5)
ASA grade I	70 (15.8)
ASA grade II	187 (42.3)
ASA grade III	171 (38.7)
ASA grade IV	14 (3.2)

Values are given as a median (range) or a number (%). ASA = American Society of Anesthesiology physical status classification system.

**Table 2 children-09-00998-t002:** Performed procedures.

Painful	459 (58.4)	Nonpainful	325 (41.4)
Bone marrow aspiration	265 (33.8)	MRI scanning	239 (30.5)
Liver biopsy	88 (11.2)	MIBG scintigraphy	33 (4.2)
Central venous catheterization	38 (4.8)	CT scanning	19 (2.4)
Renal biopsy	38 (4.8)	Renal scintigraphy	15 (1.9)
Gastrointestinal endoscopy	11 (1.4)	Other	12 (1.5)
Lumbar puncture	10 (1.3)	Auditory brainstem response	7 (0.9)
Respiratory tract endoscopy	9 (1.1)		

**Table 3 children-09-00998-t003:** Adverse events during sedation.

Minor Adverse Events		Serious Adverse Events	
Apnea	13 (1.7)	Desaturation <90% for >30 s	6 (0.8)
Airway obstruction	8 (1.0)	Thorax rigidity	1 (0.1)
IV-related complications	8 (1.0)		
Inadequate sedation/movements	5 (0.6)		
Agitation/delirium	2 (0.3)		
Coughing	2 (0.3)		
Hypersalivation	2 (0.3)		
Rash	1 (0.1)		
Bradycardia	1 (0.1)		
Bronchospasm	1 (0.1)		
Paradoxical reaction	1 (0.1)		

**Table 4 children-09-00998-t004:** Interventions during sedation.

Bag-mask ventilation	17 (2.2)
Nasopharyngeal airway	11 (1.4)
Suction	9 (1.1)
Jaw thrust	5 (0.6)
Benzodiazepines	3 (0.4)
Laryngeal mask	3 (0.4)
New IV access	2 (0.3)
Inhalational sedation	2 (0.3)
Repositioning	2 (0.3)
Endotracheal tube	1 (0.1)
Inhalation	1 (0.1)

**Table 5 children-09-00998-t005:** Bivariate regression analysis for the risk of adverse events.

Variable	Odds Ratio (95% CI)	*p*
Age (years)		
≤1	Reference	
1–6	0.303 (0.105–0.875)	0.027
>6	0.200 (0.064–0.624)	0.006
Sex		
Female	Reference	
Male	0.348 (0.172–0.704)	0.003
Sedative agent		
Propofol	Reference	
Midazolam	0.286 (0.077–1.055)	0.060
None	2.556 (0.749–8.716)	0.134
Dose of the sedative agent		
Propofol bolus	1.312 (1.128–1.526)	<0.001
Propofol continuous infusion	1.352 (0.913–2.001)	0.133
Midazolam	1.040 (0.007–152.548)	0.988
Analgesic		
Remifentanil	Reference	
Esketamine	0.401 (0.043–3.773)	0.424
Primary diagnosis		
Other	Reference	
Hematology/oncology	0.420 (0.158–1.118)	0.083
Nephrology	0.231 (0.040–1.345)	0.103
Hepatology	0.282 (0.070–1.142)	0.076
Neurology	0.748 (0.256–2.192)	0.597
Quarter of date of procedure	0.697 (0.520–0.935)	0.016
ASA grade		
I	Reference	
II	1.077 (0.335–3.463)	0.900
III	1.125 (0.363–3.491)	0.838
IV	3.253 (0.567–18.659)	0.186
Upper respiratory tract infection		
No	Reference	
Yes	1.610 (0.479–5.403)	0.441

## Data Availability

The data presented in this study are available on request from the corresponding author.

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
