# Peer review of "The Impact of a Dedicated Sedation Team on the Incidence of Complications in Pediatric Procedural Analgosedation"

_children, 2022, doi:10.3390/children9070998_

Round 1

Reviewer 1 Report

Apostolidou et al. present data that outline risk factors for adverse events in pedriatric analgosedation procedures. They collected these data in their specialized Center of Anesthesiology and Intensive Care Medicine at Hamburg-Eppendorf Hospital. The circumstances and variables leading to incidents during sedation are reasonably classified. Methodology is sound and purposeful.

My concern is that the authors do not question the generalizability of their results for other specialized anesthesiology units. The performance of their particular unit is a clear success and consequence of their well-trained personnel. Moreover, the autors highlight the constant progress of the specialized team over the course of one year. Nevertheless, the reader might be interested in a reference point for the number of adverse events and their main concomitants emerging in hospitals without a „dedicated interdisciplinary sedation team“.

Author Response

  • "My concern is that the authors do not question the generalizability of their results for other specialized anesthesiology units." For years now, we are discussing with a lot of colleagues from all over Germany the feasability of implementing such a specialized and interdisciplinary team in a tertiary center. In the most german medical centers, the service "procedural sedation" is mainly performed by either anesthesiologists or pediatric intensivists (if available) alongside their main work load, without structured planning and without reserving the personal resource for this service. Several attempts to establish such a team in other hospitals, as we are informed, failed mainly because of limited human or financial resources. With this paper, we would like to encourage more medical centers to establish such a team, in spite of commercial concerns, showing our big success on behalf of the pediatric patients.
  • "Nevertheless, the reader might be interested in a reference point for the number of adverse events and their main concomitants emerging in hospitals without a „dedicated interdisciplinary sedation team“." Unfortunately, there is no data available about the pediatric analgosedation done in the past, without a specialized team performing it. Our database starts with the first day of implementation ot the CAST. The same applys for other hospitals, where we approximately know how procedural sedation ist performed but we do not have any information about the success of this service and especially about the incidence of adverse events. Thus, we could not demonstrate the obvious improvement. You are very right, we ourselves would be very eager to have such data because we are very sure that there is a significant difference in outcome concerning this specialized service.

Reviewer 2 Report

This is a simple study design to look at a complex question.

First, we have no idea what the baseline rate of complications were BEFORE this team was implemented. How can we tell if there is improvement when we have no idea where you started?

Next, while it makes sense that you see improvement over time with the CAST team in place, the causes of risk you mention (age, prop dose, etc) may only be part of the story. The medications you list for the Out of OR procedures are the same as used in the OR, and you dont make any comparison to those procedures either. Again, there is no real way for the reader to understand whether there is improvement if the baseline is unknown. 

This paper has merit, but these lack of "controls" are a glaring hole and decrease its validity and sophistication. Please address to be considered for publication. 

Detailed issues:

The Abstract and the 1st paragraph of the "Reuslts" are either contradictory or confusing. In the abstract, line 24 states that 7.4% of children were <1yo, but then on p3 in the results you state that 12.2% of children were <1yr old. I understand that one may be referring to unique children, and the other to sedation episodes, but this is extremely confusing and you need to choose to list one or the other. 

Similarly, in the abstract you state the % of children who were ASA III/IV is very different than what you state in the Results section. Again, this may be because in one place you are referring to unique patients but the other is discrete episodes of sedation. Its just too confusing here too. 

Author Response

  • "First, we have no idea what the baseline rate of complications were BEFORE this team was implemented. How can we tell if there is improvement when we have no idea where you started?" Yes, you are right, we do not have any information about the incidence of adverse events during a procedural sedation before establishing the CAST. What we know for sure is that in the past pediatric procedural sedation in our hospital  was performed like this: 1) The pediatrician performing the sedation was the same as for the procedure itself, thus distracted both by the analgosedation during the procedure and by the procedure during the analgosedation. 2) The sedation performed by the pediatrician, thus not the specialist for this service, was performed ineffectual because "half-hearted", trying to minimize any potential anesthesiologic risk. 3) In case of adverse events needing intervention, e.g. mask bag ventilation, there was no skilled expert team other than the anesthesiologic emergency call available to control such adverse events, leading to a significant risk factor. We described the improvement within the implementation and the first year of operating of the CAST, but we did not assume that before there were more AE´s, because we do not have the data. Our main aim is to demonstrate that a specialized dedicated team can keep adverse events on a low level especially due to experience benefit, and that a sedation performed as a single person alongside the rest of the daily workload without an experienced team by the side can increase risks of inefficacy and safety.
  • "Next, while it makes sense that you see improvement over time with the CAST team in place, the causes of risk you mention (age, prop dose, etc) may only be part of the story. The medications you list for the Out of OR procedures are the same as used in the OR, and you dont make any comparison to those procedures either. Again, there is no real way for the reader to understand whether there is improvement if the baseline is unknown." Our aim was not to compare processes and anesthesiologic methods in or outside the operating room. Of course the anesthesiologist would use the same medication for the same sedation, independently if he/she is performing in- or outside the operating room. The benefit for the patient is that a lot of sedations which were impossible in the past because of lack of institutionalized structures and qualified teams can now be performed outside the operating room with minimizing the risk of adverse events and maximizing the effectivity and safety of the procedure. The impact of the CAST throughout the whole anesthesiologic department at the UKE was so huge that a lot of procedures formerly done in general anesthesia changed to analgosedation with huge success and patient satisfaction. Unfortunately we do not uprised data for this question, that is why I cannot respond to the very obvious, because every-day in the CAST seen, but not evidence-based improvement.
  • "The Abstract and the 1st paragraph of the "Reuslts" are either contradictory or confusing. In the abstract, line 24 states that 7.4% of children were <1yo, but then on p3 in the results you state that 12.2% of children were <1yr old. I understand that one may be referring to unique children, and the other to sedation episodes, but this is extremely confusing and you need to choose to list one or the other." That is correct, I apologize for this confusion, as you correctly mentioned the gap of the two percentage develops due to absolute number of infants and number of sedation of infants (lot of patients are listed with more than one sedation). The manuscript is corrected taking the number obtained from the patients and not the sedations, thus the 12,2 % is correct and transferred to the abstract.
  • "Similarly, in the abstract you state the % of children who were ASA III/IV is very different than what you state in the Results section. Again, this may be because in one place you are referring to unique patients but the other is discrete episodes of sedation. Its just too confusing here too." You are right as well, the same reason as above, the abstract is changed, sorry for the confusion.

Round 2

Reviewer 2 Report

Thank you for clarifying the numbers as noted above. 

Regarding the "baseline" levels of adverse events - i understand that you did not design the database to capture events before the CAST was implemented, but I reject the notion that you cannot look back in the medical records to determine at least some sort of information about the time-period before the CAST was "officially started."

While it might require some effort, obviously, but I imagine there HAS to be a way to determine how frequently there were complications with sedation outside of this new system since surely they were documented somewhere. I think this paper NEEDS some context to be accepted, otherwise the generalizability is totally unknown. 

Author Response

  • “Regarding the "baseline" levels of adverse events - i understand that you did not design the database to capture events before the CAST was implemented, but I reject the notion that you cannot look back in the medical records to determine at least some sort of information about the time-period before the CAST was "officially started."” As a matter of fact, there is really no concrete documentation about the procedural sedation existing in our university hospital, because the consultant intensivist or anesthesiologist was called “on demand” without even any premedication visit nor any specific preparation. Nowadays unimaginable, but unfortunately true.
  • “While it might require some effort, obviously, but I imagine there HAS to be a way to determine how frequently there were complications with sedation outside of this new system since surely they were documented somewhere. I think this paper NEEDS some context to be accepted, otherwise the generalizability is totally unknown.” The only “context” I can provide unfortunately, if you are agreeable to it, is to extend the part “discussion” with an additional passage at the beginning pointing out that in the past there was no premedication, no risk assessment, no organization, no documentation and no sufficient monitoring for procedural sedation, nowadays mainstay of the quality management system. Because of these growing demands, we implemented this CAST, with the results here in this paper presented.

If you agree, I would then change the manuscript, beginning of Discussion, and point out the background of the CAST and what I proposed above.

Thanks in advance